# Toxic Evaluation of *Cymbopogon citratus* Chemical Fractions in *E. coli*

**Fabiana Fuentes-León [1,\*], Maribel González-Pumariega [1], Marioly Vernhes Tamayo [2], Carlos Frederico Martin Menck [3] and Ángel Sánchez-Lamar [1,\*]**

[1]   Plant Biology Department, Faculty of Biology, University of Havana, Havana 10400, Cuba; bel@nauta.cu
[2]   Radiobiology Department, Center of Applied Technology and Nuclear Development (CEADEN), Havana 10400, Cuba; mariolys@ceaden.edu.cu
[3]   Microbiology Department, Instituto de Ciências Biomédicas, Universidade de São Paulo, São Paulo 05508-900, Brazil; cfmmenck@usp.br
**\***   Correspondence: fabiana@fbio.uh.cu (F.F.-L.); alamar@fbio.uh.cu (A.S.-L.); Tel.: +53-7832-8542 (F.F.-L. & A.S.-L.)

**Abstract:** *Cymbopogon citratus* (DC) Stapf is consumed as a popular decoction owing to its nice flavor and hypotensor property. Its aqueous extract radioprotector and antimutagenic properties have been experimentally demonstrated. In addition, its DNA protective activity against UV light has been proved in plasmid DNA and bacterial models. The fractioning process is important in order to identify phytocompounds responsible for this activity. In this work, the toxicity of three fractions obtained from *Cymbopogon citratus* (essential oils, butanolic and aqueous fractions) were tested using the SOS Chromotest in *Escherichia coli*. *Cymbopogon citratus* chemical fractions possess cytotoxic properties in *E. coli* in the following order butanolic > aqueous > essentials oils. Genotoxic properties were detected in any of the fractions.

**Keywords:** essential oils; butanolic and aqueous fractions; SOS assay

## 1. Introduction

The therapeutic use of plants is part of universal human culture. Over the last few years, there has been an increase in the number of studies reporting medicinal plants and dietary components as chemopreventive agents. Particularly, the usefulness of plants as sources of photochemopreventive components has been demonstrated [1]. For the last 10 years, our research group has been evaluating plant extracts against DNA damage induced by UV light [2–8]. The study of photoprotective properties of phytochemicals enriches and supports the continuous development of the pharmaceutical and cosmetic industries. To achieve this purpose, the assessment of plant extracts' safety, as well as plant-derived products, is highly relevant [9,10].

*Cymbopogon citratus* (DC) Stapf, known in Cuba as Caña Santa, is consumed as a popular decoction. This plant has been traditionally used to treat different diseases and has several pharmacological properties [11,12]. In addition, antimutagenic properties have been described for this plant, including photoprotective activity evaluated in plasmid DNA, bacterial cells, and eukaryotic organisms [4–6,13,14]. Plenty of these properties have been assayed in total extracts or decoctions, although less of them have been tested in fractioned extracts. The fractioning process is important to identify the compounds responsible for photoprotective activity. Also, the assessment of their safety is a prerequisite in antigenotoxic evaluations.

SOS Chromotest has been used for several years to test different mutagenic agents [15,16]. More recently, it has been very useful in evaluating plant extracts' genotoxicity and DNA protection [17].

In this paper, using a fluorescent protocol of SOS Chromotest in *E. coli* [18], we studied the toxic activity of essential oils, as well as butanolic and aqueous fractions of *C. citratus.*

## 2. Materials and Methods

### 2.1. Chemical Fractioning of Cymbopogon citratus Total Extract

Leaves of Caña Santa were obtained from adult plants grown in Boyeros, Havana, Cuba. The specimens were verified at the Medicinal Plants Station in Güira de Melena, Artemisa, Cuba (herbarium # 4593) [19]. Fresh leaves were triturated and boiled in distilled water (*w/v*) for 30 min. Essential oils were collected by the steam distillation method [20]. The remaining solution was fractionated by extracting in butanol (Sigma, St. Louis, MO, USA) [21]. The resulting aqueous phases were dried and vaporized to obtain butanolic and aqueous fractions. The stock solutions of oil and butanolic fraction (16 mg/mL) were diluted in Dimethyl Sulfoxide (DMSO) (Merck, Darmstadt, Germany) at 2.0 and 2.8%, respectively, and miliQ water. The aqueous fraction was diluted only in miliQ water. The concentrations evaluated were 0.1, 0.5, 1.0, 2.0, and 4.0 mg/mL.

### 2.2. Bacterial Strains and Culture

*E. coli* PQ37 strain genotype (F thr leu his-4 pyrD thi gal; K o galT lac ΔU 169 sr/300::Tn10 rpoB rpsL uvrA trp::muc+ sfiA::mud(ap,lac) cts) was used in the SOS Chromotest. The cells were grown at 37 °C under constantshaking (100 rev/min) in Luria-Bertani (LB) (Sigma) media supplemented with Ampicillin (Sigma) (25 µg/mL) until an optic densitometry (OD) of 0.4 at 600 nm.

### 2.3. SOS Chromotest

The fluorescent SOS assays described by Cuétara, et al. [18] was used to test the toxicity of the chemical fractions. Briefly, exponential phase cultures ($OD_{600nm}$ = 0.4) were 10-fold diluted in fresh LB medium (2X) supplemented with Ampicillin 25 µg/mL and dispensed in 1.5-mL tubes containing the fraction to be tested. The cells were exposed for 30 min at 4 °C. Later, cells were incubated for 2 h at 37 °C under constant shaking (100 rpm/min), and finally the enzymatic reactions were conducted. The cytotoxicity criterion was the reduction of the alkaline phosphatase expression. For the genotoxicity test, the criterion was the increase of the induction factor of the SOS response (SOSIF) [15], calculated as follows:

$$\text{SOSIF} = \frac{(\beta\text{galactosidase}/\text{alkaline phosphatase})_{\text{treated cells}}}{(\beta\text{galactosidase}/\text{alkaline phosphatase})_{\text{non}-\text{treated cells}}} \tag{1}$$

Cells harvested in medium without irradiation and irradiated were used as negative and positive controls, referred as 0.0 concentration and UVC (ultraviolet light, band C), respectively. After 30 min of incubation at 4 °C, a 1.5-mL batch was irradiated in Petri dishes with a diameter of 3 cm. UVC irradiation (λ = 254 nm, E = 45 J/m$^2$) was carried out using a Vilber Lourmat Lamp T15M 15 W (Vilber Lourmat, Suebia, Germany) at room temperature. Afterwards, cells were collected by centrifugation, resuspended in their respective treatment, and then incubated for 2 h at 30 °C under constant shaking (100 rpm/min). All measurements were taken in triplicate.

### 2.4. Statistical Analysis

The means and standard deviation of alkaline phosphatase and SOSIF were determined. Controls and treatments were analyzed using the Kolmogorov-Smirnov test. Variance homogeneity (Brown Forsythe test) and single classification ANOVA were also conducted. Values for different treatments were compared with the negative control using a Dunett test ($p < 0.05$), according to STATISTICA 6.0.

## 3. Results

The toxicity of chemical fractions of *C. citratus* to *E. coli* PQ37 cells was investigated. The alkaline phosphatase assay in treated cells (protein synthesis inhibition indices) was used as the cytotoxicity criterion. The capability to produce primary DNA damage by the plant fractions was determined measuring the induction of SOS genes. It is known that a compound is classified as not genotoxic if the SOSIF remains ≤1.5, not conclusive if SOSIF is between 1.5 and 2.0, and genotoxic if SOSIF exceeds 2.0 [22].

The results showed that the essential oils (4.0 mg/mL), aqueous fraction (1.0–2.0 mg/mL), and butanolic fraction (0.5–4.0 mg/mL) decrease the alkaline phosphatase constitutive expression when they are compared to non-treated cells (0.0 mg/mL). Any fraction induced DNA damage at the concentrations tested according to the criterion described by Kevekores et al. [22] (Table 1).

**Table 1.** Effects of essential oils, and aqueous and butanolic fractions from *Cymbopogon citratus* on the alkaline phopatase (AP) expression and induction of SOS phenomenon (SOSIF) in *E. coli* cells.

| Concentrations (mg/mL) | Chemical Fractions | | | | | |
| --- | --- | --- | --- | --- | --- | --- |
| | Essential Oils | | Aqueous | | Butanolic | |
| | AP | SOSIF | AP | SOSIF | AP | SOSIF |
| 0.0 | 6.34 ± 0.13 | 1.00 ± 0.05 | 3.73 ± 0.16 | 1.02 ± 0.02 | 3.98 ± 0.32 | 1.02 ± 0.05 |
| 0.1 | 4.87 ± 0.28 | 0.98 ± 0.09 | 3.69 ± 0.13 | 1.02 ± 0.03 | 3.66 ± 0.26 | 1.00 ± 0.06 |
| 0.5 | 6.67 ± 0.96 | 0.88 ± 0.12 | 3.40 ± 0.14 | 1.01 ± 0.03 | 3.15 ± 0.11 * | 1.00 ± 0.03 |
| 1.0 | 6.79 ± 0.68 | 0.94 ± 0.09 | 3.43 ± 0.19 | 1.11± 0.04 | 2.91 ± 0.29 * | 0.93 ± 0.02 |
| 2.0 | 5.00 ± 0.15 | 0.80 ± 0.05 | 3.21 ± 0.14 * | 1.12 ± 0.04 | 2.74 ± 0.05 * | 1.01 ± 0.05 |
| 4.0 | 3.68 ± 0.12 * | 0.90 ± 0.11 | 2.94 ± 0.07 * | 1.34 ± 0.12 * | 2.75 ± 0.03 * | 1.12 ± 0.07 |
| UVC (5 J/m$^2$) | 4.32 ± 0.09 * | 7.85 ± 0.84 * | 3.11 ± 0.13 * | 7.69 ± 0.74 * | 2.15 ± 0.23 * | 6.55 ± 0.30 * |

(*) significant in Dunett Test $p < 0.05$.

## 4. Discussion

In our work, the fractions obtained from *C. citratus* were cytotoxic in *E. coli* cells PQ37 strain. The butanolic fraction was the most cytotoxic. The essentials oils and aqueous fraction were toxic at the highest concentrations tested. On the other hand, any fractions caused damage in bacterial DNA. In previous experiments in *Caulobacter crescentus* model, the essential oils of *C. citratus* also showed cytotoxic effect at 4.0 mg/mL [6]. Interestingly, in contrast with our results here, the aqueous fraction did not show a toxic result at any concentration tested, and the butanolic fraction showed a strongly genotoxic result [6]. Probably the genetic background of *C. crescentus* and *E. coli* used in our studies could be related to the differences observed. Furthermore, transcriptional fusion in *E. coli* PQ37 and *C. crescentus* NA1000 pp3213 could reinforce this hypothesis [15,16,23].

Using different extraction methods, it was demonstrated that the aqueous extract of *Cymbopogon citratus* did not induce DNA primary in the plasmidic model, and in *E. coli* PQ-37 cells reduced alkaline phosphates levels at 4.0 mg/mL. In this bacterial cell, concentrations higher than 2.0 mg/mL were genotoxic [4,5,18,23].

It is important to remark that in the fractioning process, the phytocompounds of the total extract are separated and concentrated. These aspects could explain the increase in toxicity found in chemical fractions in contrast to the total extract when the same concentrations are tested.

The results obtained here complement previous evaluations of the total extract and chemical fractions [4–6]. Although complementary tests, such as in vitro human epidermis models, should be conducted in order to guarantee the safety of these fractions, these results are an important contribution to the future studies of *Cymbopogon citratus* fractions as chemopreventive agents. Furthermore, nontoxic concentrations bring a good perspective to their potential in the pharmaceutical and cosmetic industries.

## 5. Conclusions

The essential oils, aqueous and butanolic fractions obtained from *Cymbopogon citrates* possess cytotoxic effects in *E. coli* cells. However, non genotoxic effects were detected at the assayed concentrations.

**Acknowledgments:** CAPES (Brazil)-MES (Cuba) collaborative project financed this work.

**Author Contributions:** Ángel Sánchez-Lamar, Carlos Frederico Martin Menck and Maribel González-Pumariega conceived and designed the experiments; Maribel González-Pumariega and Marioly Vernhes Tamayo performed the experiments; Fabiana Fuentes-León analyzed the data; Ángel Sánchez-Lamar, Carlos Frederico Martin Menck and Marioly Vernhes Tamayo contributed reagents/materials/analysis tools; Fabiana Fuentes-León wrote the paper.

**Conflicts of Interest:** The authors declare no conflict of interest.

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
