# Peer review of "Toxic Evaluation of Cymbopogon citratus Chemical Fractions in E. coli"

_cosmetics, doi:10.3390/cosmetics4020020_

Round 1

Reviewer 1 Report

This short communication addresses the toxicity of three fractions of Cymbopogon citratus using a commercial test to evaluate the genotoxic potential of chemicals in Escherichia coli.

The introduction is short and should contain more background information previously published, although the key points are stated, but too much summarised. As an example, more information regarding the SOS Chromotest should be introduced as this test has been used for several years and several chemical agents evaluated with it.

The material and methods section is also very short, especially in the SOS Chromotest, as the authors state in the introduction (page 1, lines 41-43: “In this paper, using a modified protocol of SOS Chromotest in E. coli, we studied the toxic activity of essential oils, butanolic and aqueous fractions of C. citratus.”) that used this test with modifications, this is not explicit enough. Thus, at least this section should be more detailed.

Also here in the material and methods section information regarding the manufacturers of the chemicals, broth medium (e.g. DMSO, LB media, Ampicillin) is lacking. Additionally, here an abbreviation appears for the first time (FI in page 2, line 66) with no previous definition. As stated in the instruction for authors: “Abbreviations should be defined in parentheses the first time they appear in the abstract, main text, and in figure or table captions.”

The table should be included in the result section and not in the discussion as authors performed. Also here, in the results section, should be explained how the authors interpreted the cytotoxicity by the alkaline phosphatase assay, because no cut off values are shown to be able to interpret the results. Regarding the genotoxic assay, looking at the table, none of the fractions induces genotoxicity, or am I interpreting it wrongly? Where are the results that the authors state? What is UVC? This section must be completely rewritten to be clearer to understand. Also, complementary tests should be included in the study, such as, in vitro reconstructed human epidermis models.

The discussion section should be reformulated, as authors just affirm results with poor discussion of results, especially considering that no information regarding cut off values of cytotoxicity is shown to be able to interpret and even understand the “discussion” and conclusions of authors. Regarding genotoxicity the conclusions also lack information to be able to understand authors ideas and interpretations. Some information just appear without any connection or association with the results reported in this study. Again, here in this section authors use abbreviations without previous definition (page 2, line 84; page 3, line 95). Authors just state Ames test without further discussion. Authors use word reductions that are inadmissible in scientific communications (page 3, line 96 – didn’t instead of did not).  

For all these reasons the paper in its present form is not acceptable for publication. Major modifications should be implemented in order to clarify the methods, results and discussion sections.

Furthermore, it is strongly recommended that the manuscript should be previously corrected by an English speaking person before further submission. Also, some phrases structures are incorrect and confusing. 

Author Response

Thanks for your suggestion about our short communication tittled: Toxic  evaluation of Cymbopogon citratus chemical fractions in E. coli.

In order to respond the reviewer comments some modification were made in document. We are sending a summary with the changes

In the introduction

§       A paragraph regarding SOS Chromotest assay and other ideas were added in other to increase the comprehension.

In materials and methods section

§        The fluorescent protocol proposed (Cuétara et al., 2012) was briefly described.

§        Positive and negative controls were declared as 0.0 mg/mL of extracts concentration with and without UVC irradiation, respectively.

§        The abbreviation FI was changed by SOSIF.

§        Chemical factures were added for DMSO, LB, Ampicillin and butanol.

 In the results

§      The table 1 was moved to this section.

All the fractions treatments were compared with their corresponding negative control (0.0 mg/mL). In case of cytotoxic evaluation, a significant reduction of  alkaline phosphatase enzyme was taken as cytotoxic criterion.

For genotoxicity test, the increase of SOSIF value was the taken criterion. According to Kevebordes and colaborators. Mutation Research 445 (1999) 81–91. A compound is classified as ‘‘not genotoxic’’ if the induction factor is minor than 1.5, as ‘‘marginal’’ if the induction factor is between 1.5 and 2.0, and as ‘‘genotoxic’’ if the induction factor exceeds 2.0. According to this any of the fractions caused damage in DNA even if the higher concentration tested (4.0 mg/mL) for aqueous fraction significantly increases SOSIF.

In discussion

§      The redaction of text was modified.

The aim of this paper was to evaluate the toxic effects of three chemical fractions extracted from Cymbopogon citratus. We agree that further studies in different experimental models are required. However, our results are just a prerequisite required for evaluated the photoprotection of these fractions in bacterial models.

Finally, English corrections were made.

We are attaching you a modified manuscript

Thank you for your time

Reviewer 2 Report

change ...toxic in genotoxic

Author Response

Thanks for your suggestion about our short communication tittled: Toxic  evaluation of Cymbopogon citratus chemical fractions in E. coli.

In order to respond the reviewer comments some modification were made in document. We are sending a summary with the changes

In titled we prefer the term toxic, because of we evaluate the cytotoxic and genotoxic effect of fractions in E. coli cells.

The methods were modified in order to improved the described the experimental conditions.

The results and the conclusions were lightly modified.

In the discussion section, the text was modified.

Finally, English corrections were made.

We are attaching you a modified manuscript

Thank you for your time

Round 2

Reviewer 1 Report

The authors have corrected the paper accordingly to the suggested.

Some minor English phrase corrections should be performed, eg:

-          Page 1, line 22 and 23 – Genotoxic properties were detected in any of the fractions.

Information of the city and country of the manufacturers should be introduced, eg: (Merck,Darmstadt,Germany)...